# ENERGY-BASED SPHERICAL SPARSE CODING

**Bailey Kong and Charless C. Fowlkes**
Department of Computer Science
University of California, Irvine
Irvine, CA 92697 USA
{bhkong,fowlkes}@ics.uci.edu

## ABSTRACT

In this paper, we explore an efficient variant of convolutional sparse coding with unit norm code vectors where reconstruction quality is evaluated using an inner product (cosine distance). To use these codes for discriminative classification, we describe a model we term Energy-Based Spherical Sparse Coding (EB-SSC) in which the hypothesized class label introduces a learned linear bias into the coding step. We evaluate and visualize performance of stacking this encoder to make a deep layered model for image classification.

## 1 INTRODUCTION

Sparse coding has been widely studied as a representation for images, audio and other vectorial data. This has been a highly successful method that has found its way into many applications, from signal compression and denoising (Donoho, 2006; Elad & Aharon, 2006) to image classification (Wright et al., 2009), to modeling neuronal receptive fields in visual cortex (Olshausen & Field, 1997). Since its introduction, subsequent works have brought sparse coding into the supervised learning setting by introducing classification loss terms to the original formulation to encourage features that are not only able to reconstruct the original signal but are also discriminative (Jiang et al., 2011; Yang et al., 2010; Zeiler et al., 2010; Ji et al., 2011; Zhou et al., 2012; Zhang et al., 2013).

While supervised sparse coding methods have been shown to find more discriminative features leading to improved classification performance over their unsupervised counterparts, they have received much less attention in recent years and have been eclipsed by simpler feed-forward architectures.

This is in part because sparse coding is computationally expensive. Convex formulations of sparse coding typically consist of a minimization problem over an objective that includes a least-squares (LSQ) reconstruction error term plus a sparsity inducing regularizer.

Because there is no closed-form solution to this formulation, various iterative optimization techniques are generally used to find a solution (Zeiler et al., 2010; Bristow et al., 2013; Yang et al., 2013; Heide et al., 2015). In applications where an approximate solution suffices, there is work that learns non-linear predictors to estimate sparse codes rather than solve the objective more directly (Gregor & LeCun, 2010). The computational overhead for iterative schemes becomes quite significant when training discriminative models due to the demand of processing many training examples necessary for good performance, and so sparse coding has fallen out of favor by not being able to keep up with simpler non-iterative coding methods.

In this paper we introduce an alternate formulation of sparse coding using unit length codes and a reconstruction loss based on the cosine similarity. Optimal sparse codes in this model can be computed in a non-iterative fashion and the coding objective lends itself naturally to embedding in a discriminative, energy-based classifier which we term *energy-based spherical sparse coding (EB-SSC)*. This bi-directional coding method incorporates both top-down and bottom-up information where the features representation depends on both a hypothesized class label and the input signal. Like Cao et al. (2015), our motivation for bi-directional coding comes from the "Biased Competition Theory", which suggests that visual processing can be biased by other mental processes (e.g., top-down influence) to prioritize certain features that are most relevant to current task. Fig. 1 illustrates the flow of computation used by our SSC and EB-SSC building blocks compared to a standard feed-forward layer.

Our energy based approach for combining top-down and bottom-up information is closely tied to the ideas of Larochelle & Bengio (2008); Ji et al. (2011); Zhang et al. (2013); Li & Guo (2014)—although the model details are substantially different (e.g., Ji et al. (2011) and Zhang et al. (2013) use sigmoid non-linearities while Li & Guo (2014) use separate representations for top-down and bottom-up information). The energy function of Larochelle & Bengio (2008) is also similar but includes an extra classification term and is trained as a restricted Boltzmann machine.

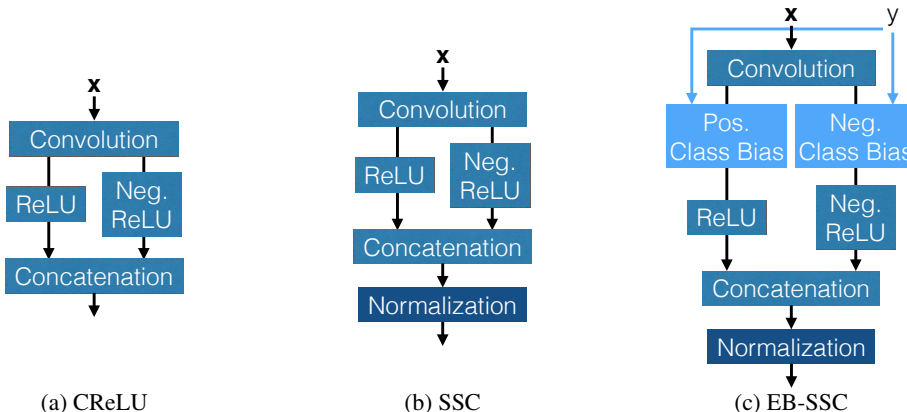

|           (a) CReLU           |           (b) SSC           |           (c) EB-SSC           |

Figure 1: Building blocks for coding networks explored in this paper. Our coding model uses non-linearities that are closely related to the standard ReLU activation function. (a) Keeping both positive and negative activations provides a baseline feed-forward model termed concatenated ReLU (CReLU). (b) Our spherical sparse coding layer has a similar structure but with an extra bias and normalization step. Our proposed energy-based model uses (c) energy-based spherical sparse coding (EB-SSC) blocks that produces sparse activations which are not only positive and negative, but are class-specific. These blocks can be stacked to build deeper architectures.

## 1.1 NOTATION

Matrices are denoted as uppercase bold (e.g., $\mathbf{A}$), vectors are lowercase bold (e.g., $\mathbf{a}$), and scalars are lowercase (e.g., $a$). We denote the transpose operator with $^\mathsf{T}$, the element-wise multiplication operator with $\odot$, the convolution operator with $*$, and the cross-correlation operator with $\star$. For vectors where we dropped the subscript $k$ (e.g., $\mathbf{d}$ and $\mathbf{z}$), we refer to a super vector with $K$ components stacked together (e.g., $\mathbf{z} = [\mathbf{z}_1^\mathsf{T}, \ldots, \mathbf{z}_K^\mathsf{T}]^\mathsf{T}$).

## 2 ENERGY-BASED SPHERICAL SPARSE CODING

Energy-based models capture dependencies between variables using an energy function that measure the compatibility of the configuration of variables (LeCun et al., 2006). To measure the compatibility between the top-down and bottom-up information, we define the energy function of EB-SSC to be the sum of bottom-up coding term and a top-down classification term:

$$E(\mathbf{x}, y, \mathbf{z}) = E_{code}(\mathbf{x}, \mathbf{z}) + E_{class}(y, \mathbf{z}). \tag{1}$$

The bottom-up information (input signal $\mathbf{x}$) and the top-down information (class label $y$) are tied together by a latent feature map $\mathbf{z}$.

## 2.1 BOTTOM-UP RECONSTRUCTION

To measure the compatibility between the input signal $\mathbf{x}$ and the latent feature maps $\mathbf{z}$, we introduce a novel variant of sparse coding that is amenable to efficient feed-forward optimization. While the idea behind this variant can be applied to either patch-based or convolutional sparse coding, we specifically use the convolutional variant that shares the burden of coding an image among nearby overlapping dictionary elements. Using such a shift-invariant approach avoids the need to learn dictionary elements which are simply translated copies of each other, freeing up resources to discover more diverse and specific filters (see Kavukcuoglu et al. (2010)).

Convolutional sparse coding (CSC) attempts to find a set of dictionary elements $\{\mathbf{d}_1, \dots, \mathbf{d}_K\}$ and corresponding sparse codes $\{\mathbf{z}_1, \dots, \mathbf{z}_K\}$ so that the resulting reconstruction, $\mathbf{r} = \sum_{k=1}^{K} \mathbf{d}_k * \mathbf{z}_k$ accurately represents the input signal $\mathbf{x}$. This is traditionally framed as a least-squares minimization with a sparsity inducing prior on $\mathbf{z}$:

$$\arg\min_{\mathbf{z}} \|\mathbf{x} - \sum_{k=1}^{K} \mathbf{d}_k * \mathbf{z}_k\|_2^2 + \beta \|\mathbf{z}\|_1. \tag{2}$$

Unlike standard feed-forward CNN models that convolve the input signal $\mathbf{x}$ with the filters, this energy function corresponds to a generative model where the latent feature maps $\{\mathbf{z}_1, \dots, \mathbf{z}_K\}$ are convolved with the filters and compared to the input signal (Bristow et al., 2013; Heide et al., 2015; Zeiler et al., 2010).

To motivate our novel variant of CSC, consider expanding the squared reconstruction error $\|\mathbf{x} - \mathbf{r}\|_2^2 = \|\mathbf{x}\|_2^2 - 2\mathbf{x}^\mathsf{T}\mathbf{r} + \|\mathbf{r}\|_2^2$. If we constrain the reconstruction $\mathbf{r}$ to have unit norm, the reconstruction error depends entirely on the inner product between $\mathbf{x}$ and $\mathbf{r}$ and is equivalent to the cosine similarity (up to additive and multiplicative constants). This suggests the closely related *unit-length* reconstruction problem:

$$\arg\max_{\mathbf{z}} \mathbf{x}^\mathsf{T}\big(\sum_{k=1}^{K} \mathbf{d}_k * \mathbf{z}_k\big) - \beta \|\mathbf{z}\|_1 \tag{3}$$

$$\text{s.t. } \big\|\sum_{k=1}^{K} \mathbf{d}_k * \mathbf{z}_k\big\|_2 \leq 1$$

In Appendix A we show that, given an optimal unit length reconstruction $\bar{\mathbf{r}}^*$ with corresponding codes $\bar{\mathbf{z}}^*$, the solution to the least squares reconstruction problem (Eq. 2) can be computed by a simple scaling $\mathbf{r}^* = (\mathbf{x}^\mathsf{T}\bar{\mathbf{r}}^* - \frac{\beta}{2}\|\bar{\mathbf{z}}^*\|_1)\bar{\mathbf{r}}^*$.

The unit-length reconstruction problem is no easier than the original least-squares optimization due to the constraint on the reconstruction which couples the codes for different filters. Instead consider a simplified constraint on $\mathbf{z}$ which we refer to as *spherical sparse coding (SSC)*:

$$\arg\max_{\|\mathbf{z}\|_2 \leq 1} E_{code}(\mathbf{x}, \mathbf{z}) = \arg\max_{\|\mathbf{z}\|_2 \leq 1} \mathbf{x}^\mathsf{T}\big(\sum_{k=1}^{K} \mathbf{d}_k * \mathbf{z}_k\big) - \beta \|\mathbf{z}\|_1. \tag{4}$$

In 2.3 below, we show that the solution to this problem can be found very efficiently without requiring iterative optimization.

This problem is a relaxation of convolutional sparse coding since it ignores non-orthogonal interactions between the dictionary elements[1]. Alternately, assuming unit norm dictionary elements, the code norm constraint can be used to upper-bound the reconstruction length. We have by the triangle and Young's inequality that:

$$\big\|\sum_k \mathbf{d}_k * \mathbf{z}_k\big\|_2 \leq \sum_k \|\mathbf{d}_k * \mathbf{z}_k\|_2 \leq \sum_k \|\mathbf{d}_k\|_1 \|\mathbf{z}_k\|_1 \leq D \sum_k \|\mathbf{z}_k\|_2 \tag{5}$$

where the factor $D$ is the dimension of $\mathbf{z}_k$ and arises from switching from the 1-norm to the 2-norm. Since $D \sum_k \|\mathbf{z}_k\|_2 \leq 1$ is a tighter constraint we have

$$\max_{\|\sum_k \mathbf{d}_k * \mathbf{z}_k\|_2 \leq 1} E_{code}(\mathbf{x}, \mathbf{z}) \geq \max_{\sum_k \|\mathbf{z}_k\|_2 \leq \frac{1}{D}} E_{code}(\mathbf{x}, \mathbf{z}) \tag{6}$$

However, this relaxation is very loose, primarily due to the triangle inequality. Except in special cases (e.g., if the dictionary elements have disjoint spectra) the SSC codes will be quite different from the standard least-squares reconstruction.

---

[1] We note that our formulation is also closely related to the dynamical model suggested by Rozell et al. (2008), but without the dictionary-dependent lateral inhibition between feature maps. Lateral inhibition can solve the unit-length reconstruction formulation of standard sparse coding but requires iterative optimization.

## 2.2 TOP-DOWN CLASSIFICATION

To measure the compatibility between the class label $y$ and the latent feature maps $\mathbf{z}$, we use a set of one-vs-all linear classifiers. To provide more flexibility, we generalize this by splitting the code vector into positive and negative components:

$$\mathbf{z}_k = \mathbf{z}_k^+ + \mathbf{z}_k^- \quad \mathbf{z}_k^+ \geq 0 \quad \mathbf{z}_k^- \leq 0$$

and allow the linear classifier to operate on each component separately. We express the classifier score for a hypothesized class label $y$ by:

$$E_{class}(y, \mathbf{z}) = \sum_{k=1}^{K} \mathbf{w}_y^{+\mathsf{T}} \mathbf{z}_k^+ + \sum_{k=1}^{K} \mathbf{w}_y^{-\mathsf{T}} \mathbf{z}_k^-. \tag{7}$$

The classifier thus is parameterized by a pair of weight vectors ($\mathbf{w}_{yk}^+$ and $\mathbf{w}_{yk}^-$) for each class label $y$ and $k$-th channel of the latent feature map.

This splitting, sometimes referred to as full-wave rectification, is useful since a dictionary element and its negative do not necessarily have opposite visual semantics. This splitting also allows the classifier the flexibility to assign distinct meanings or alternately be completely invariant to contrast reversal depending on the problem domain. For example, Shang et al. (2016) found CNN models with ReLU non-linearities which discard the negative activations tend to learn pairs of filters which are related by negation. Keeping both positive and negative responses allowed them to halve the number of dictionary elements.

We note that it is also straightforward to introduce spatial average pooling prior to classification by introducing a fixed linear operator $\mathbf{P}$ used to pool the codes (e.g., $\mathbf{w}_y^{+\mathsf{T}} \mathbf{P} \mathbf{z}_k^+$). This is motivated by a variety of hand-engineered feature extractors and sparse coding models, such as Ren & Ramanan (2013), which use spatially pooled histograms of sparse codes for classification. This fixed pooling can be viewed as a form of regularization on the linear classifier which enforces shared weights over spatial blocks of the latent feature map. Splitting is also quite important to prevent information loss when performing additive pooling since positive and negative components of $\mathbf{z}_k$ can cancel each other out.

## 2.3 CODING

Bottom-up reconstruction and top-down classification each provide half of the story, coupled by the latent feature maps. For a given input $\mathbf{x}$ and hypothesized class $y$, we would like to find the optimal activations $\mathbf{z}$ that maximize the joint energy function $E(\mathbf{x}, y, \mathbf{z})$. This requires solving the following optimization:

$$\underset{\|\mathbf{z}\|_2 \leq 1}{\arg\max} \; \mathbf{x}^{\mathsf{T}} \Big( \sum_{k=1}^{K} \mathbf{d}_k * \mathbf{z}_k \Big) - \beta \|\mathbf{z}\|_1 + \sum_{k=1}^{K} \mathbf{w}_{yk}^{+\mathsf{T}} \mathbf{z}_k^+ + \sum_{k=1}^{K} \mathbf{w}_{yk}^{-\mathsf{T}} \mathbf{z}_k^-, \tag{8}$$

where $\mathbf{x} \in \mathbb{R}^D$ is an image and $y \in \mathcal{Y}$ is a class hypothesis. $\mathbf{z}_k \in \mathbb{R}^F$ is the $k$-th component latent variable being inferred; $\mathbf{z}_k^+$ and $\mathbf{z}_k^-$ are the positive and negative coefficients of $\mathbf{z}_k$, such that $\mathbf{z}_k = \mathbf{z}_k^+ + \mathbf{z}_k^-$. The parameters $\mathbf{d}_k \in \mathbb{R}^M$, $\mathbf{w}_{yk}^+ \in \mathbb{R}^F$, and $\mathbf{w}_{yk}^- \in \mathbb{R}^F$ are the dictionary filter, positive coefficient classifier, and negative coefficient classifier for the $k$-th component respectively. A key aspect of our formulation is that the optimal codes can be found very efficiently in closed-form—in a feed-forward manner (see Appendix B for a detailed argument).

### 2.3.1 ASYMMETRIC SHRINKAGE

To describe the coding processes, let us first define a generalized version of the shrinkage function commonly used in sparse coding. Our asymmetric shrinkage is parameterized by upper and lower thresholds $-\beta^- \leq \beta^+$

$$\text{shrink}_{(\beta^+, \beta^-)}(v) = \begin{cases} v - \beta^+ & \text{if } v - \beta^+ > 0 \\ 0 & \text{otherwise} \\ v + \beta^- & \text{if } v + \beta^- < 0 \end{cases} \tag{9}$$

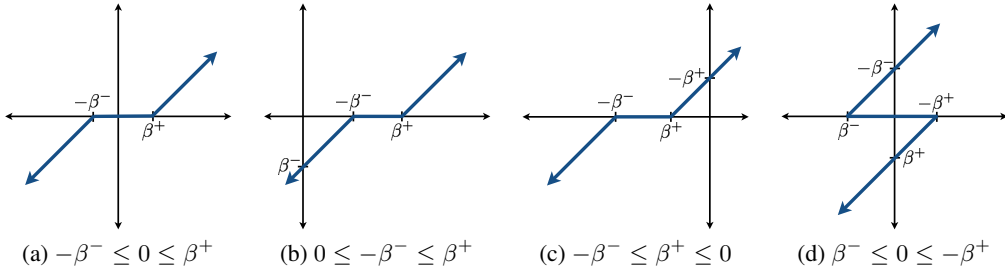

(a) $-\beta^- \leq 0 \leq \beta^+$  (b) $0 \leq -\beta^- \leq \beta^+$  (c) $-\beta^- \leq \beta^+ \leq 0$  (d) $\beta^- \leq 0 \leq -\beta^+$

Figure 2: Comparing the behavior of asymmetric shrinkage for different settings of $\beta^+$ and $\beta^-$. (a)-(c) satisfy the condition that $-\beta^- \leq \beta^+$ while (d) does not.

Fig. 2 shows a visualization of this function which generalizes the standard shrinkage proximal operator by allowing for the positive and negative thresholds. In particular, it corresponds to the proximal operator for a version of the $\ell_1$-norm that penalizes the positive and negative components with different weights $|\mathbf{v}|_{asym} = \beta^+ \|\mathbf{v}^+\|_1 + \beta^- \|\mathbf{v}^-\|_1$. The standard shrink operator corresponds to $\mathrm{shrink}_{(\beta, -\beta)}(v)$ while the rectified linear unit common in CNNs is given by a limiting case $\mathrm{shrink}_{(0, -\infty)}(v)$. We note that $-\beta^- \leq \beta^+$ is required for $\mathrm{shrink}_{(\beta^+, \beta^-)}$ to be a proper function (see Fig. 2).

### 2.3.2 FEED-FORWARD CODING

We now describe how codes can be computed in a simple feed-forward pass. Let

$$\boldsymbol{\beta}_{yk}^+ = \beta - \mathbf{w}_{yk}^+, \qquad \boldsymbol{\beta}_{yk}^- = \beta - \mathbf{w}_{yk}^- \tag{10}$$

be vectors of positive and negative biases whose entries are associated with a spatial location in the feature map $k$ for class $y$. The optimal code $\mathbf{z}$ can be computed in three sequential steps:

1. Cross-correlate the data with the filterbank $\mathbf{d}_k \star \mathbf{x}$

2. Apply an asymmetric version of the standard shrinkage operator

$$\tilde{\mathbf{z}}_k = \mathrm{shrink}_{(\boldsymbol{\beta}_{yk}^+, \boldsymbol{\beta}_{yk}^-)}(\mathbf{d}_k \star \mathbf{x}) \tag{11}$$

   where, with abuse of notation, we allow the shrinkage function (Eq. 9) to apply entries in the vectors of threshold parameter pairs $\boldsymbol{\beta}_{yk}^+, \boldsymbol{\beta}_{yk}^-$ to the corresponding elements of the argument.

3. Project onto the feasible set of unit length codes

$$\mathbf{z}^* = \frac{\tilde{\mathbf{z}}}{\|\tilde{\mathbf{z}}\|_2}. \tag{12}$$

### 2.3.3 RELATIONSHIP TO CNNS:

We note that this formulation of coding has a close connection to single layer convolutional neural network (CNN). A typical CNN layer consists of convolution with a filterbank followed by a non-linear activation such as a rectified linear unit (ReLU). ReLUs can be viewed as another way of inducing sparsity, but rather than coring the values around zero like the shrink function, ReLU truncates negative values. On the other hand, the asymmetric shrink function can be viewed as the sum of two ReLUs applied to appropriately biased inputs:

$$\mathrm{shrink}_{(\beta^+, \beta^-)}(x) = \mathrm{ReLU}(x - \beta^+) - \mathrm{ReLU}(-(x + \beta^-)),$$

SSC coding can thus be seen as a CNN in which the ReLU activation has been replaced with shrinkage followed by a global normalization.

## 3 LEARNING

We formulate supervised learning using the softmax log-loss that maximizes the energy for the true class label $y_i$ while minimizing energy of incorrect labels $\bar{y}$.

$$\underset{\mathbf{d},\mathbf{w}^+,\mathbf{w}^-,\beta \geq 0}{\arg\min} \frac{\alpha}{2}(\|\mathbf{w}^+\|_2^2 + \|\mathbf{w}^-\|_2^2 + \|\mathbf{d}\|_2^2)$$

$$+ \frac{1}{N}\sum_{i=1}^{N}[-\max_{\|\mathbf{z}\|_2 \leq 1} E(\mathbf{x}_i, y_i, \mathbf{z}) + \log \sum_{\bar{y} \in \mathcal{Y}} \max_{\|\bar{\mathbf{z}}\|_2 \leq 1} e^{E(\mathbf{x}_i, \bar{y}, \bar{\mathbf{z}})}], \tag{13}$$

$$\text{s.t.} \quad -(\beta - \mathbf{w}_{yk}^-) \leq (\beta - \mathbf{w}_{yk}^+) \quad \forall y, k$$

where $\alpha$ is the hyperparameter regularizing $\mathbf{w}_y^+$, $\mathbf{w}_y^-$, and $\mathbf{d}$. We constrain the relationship between $\beta$ and the entries of $\mathbf{w}_y^+$ and $\mathbf{w}_y^-$ in order for the asymmetric shrinkage to be a proper function (see Sec. 2.3.1 and Appendix B for details).

In classical sparse coding, it is typical to constrain the $\ell_2$-norm of each dictionary filter to unit length. Our spherical coding objective behaves similarly. For any optimal code $\mathbf{z}^*$, there is a 1-dimensional subspace of parameters for which $\mathbf{z}^*$ is optimal given by scaling $\mathbf{d}$ inversely to $\mathbf{w}$, $\beta$. For simplicity of the implementation, we opt to regularize $\mathbf{d}$ to assure a unique solution. However, as Tygert et al. (2015) point out, it may be advantageous from the perspective of optimization to explicitly constrain the norm of the filter bank.

Note that unlike classical sparse coding, where $\beta$ is a hyperparameter that is usually set using cross-validation, we treat it as a parameter of the model that is learned to maximize performance.

### 3.1 OPTIMIZATION

In order to solve Eq. 13, we explicitly formulate our model as a directed-acyclic-graph (DAG) neural network with shared weights, where the forward-pass computes the sparse code vectors and the backward-pass updates the parameter weights. We optimize the objective using stochastic gradient descent (SGD).

As mentioned in Sec. 2.3 shrinkage function is assymetric with parameters $\boldsymbol{\beta}_{yk}^+$ or $\boldsymbol{\beta}_{yk}^-$ as defined in Eq. 10. However, the inequality constraint on their relationship to keep the shrinkage function a proper function is difficult to enforce when optimizing with SGD. Instead, we introduce a central offset parameter and reduce the ordering constraint to pair of positivity constraints. Let

$$\hat{\mathbf{w}}_{yk}^+ = \boldsymbol{\beta}_{yk}^+ - b_k \qquad \hat{\mathbf{w}}_{yk}^- = \boldsymbol{\beta}_{yk}^- + b_k \tag{14}$$

be the modified linear "classifiers" relative to the central offset $b_k$. It is straightforward to see that if $\boldsymbol{\beta}_{yk}^+$ and $\boldsymbol{\beta}_{yk}^-$ that satisfy the constrain in Eq. 13, then adding the same value to both sides of the inequality will not change that. However, taking $b_k$ to be a midpoint between them, then both $\boldsymbol{\beta}_{yk}^+ - b_k$ and $\boldsymbol{\beta}_{yk}^- + b_k$ will be strictly non-negative.

Using this variable substitution, we rewrite the energy function (Eq. 1) as

$$E'(\mathbf{x}, y, \mathbf{z}) = \mathbf{x}^\mathsf{T}\Big(\sum_{k=1}^{K}\mathbf{d}_k * \mathbf{z}_k\Big) + \sum_{k=1}^{K} b_k \mathbf{1}^\mathsf{T}\mathbf{z}_k - \sum_{k=1}^{K}\hat{\mathbf{w}}_{yk}^{+\mathsf{T}}\mathbf{z}_k^+ + \sum_{k=1}^{K}\hat{\mathbf{w}}_{yk}^{-\mathsf{T}}\mathbf{z}_k^-. \tag{15}$$

where $\mathbf{b}$ is constant offset for each code channel. The modified linear "classification" terms now take on a dual role of inducing sparsity and measuring the compatibility between $\mathbf{z}$ and $y$.

This yields a modified learning objective that can easily be solved with existing implementations for learning convolutional neural nets:

$$\underset{\mathbf{d},\hat{\mathbf{w}}^+,\hat{\mathbf{w}}^-,\mathbf{b}}{\arg\min} \frac{\alpha}{2}(\|\hat{\mathbf{w}}^+\|_2^2 + \|\hat{\mathbf{w}}^-\|_2^2 + \|\mathbf{d}\|_2^2)$$

$$+ \frac{1}{N}\sum_{i=1}^{N}[-\max_{\|\mathbf{z}\|_2 \leq 1} E'(\mathbf{x}_i, y_i, \mathbf{z}) + \log \sum_{\bar{y} \in \mathcal{Y}} \max_{\|\bar{\mathbf{z}}\|_2 \leq 1} e^{E'(\mathbf{x}_i, \bar{y}, \bar{\mathbf{z}})}], \tag{16}$$

$$\text{s.t.} \quad \hat{\mathbf{w}}_{yk}^+, \hat{\mathbf{w}}_{yk}^- \succeq 0 \quad \forall y, k$$

where $\hat{\mathbf{w}}^+$ and $\hat{\mathbf{w}}^-$ are the new sparsity inducing classifiers, and $\mathbf{b}$ are the arbitrary origin points. In particular, adding the $K$ origin points allows us to enforce the constraint by simply projecting $\hat{\mathbf{w}}^+$ and $\hat{\mathbf{w}}^-$ onto the positive orthant during SGD.

### 3.1.1 STACKING BLOCKS

We also examine stacking multiple blocks of our energy function in order to build a hierarchical representation. As mentioned in Sec. 3.1.1, the optimal codes can be computed in a simple feed-forward pass—this applies to shallow versions of our model. When stacking multiple blocks of our energy-based model, solving for the optimal codes cannot be done in a feed-forward pass since the codes for different blocks are coupled (bilinearly) in the joint objective. Instead, we can proceed in an iterative manner, performing block-coordinate descent by repeatedly passing up and down the hierarchy updating the codes. In this section we investigate the trade-off between the number of passes used to find the optimal codes for the stacked model and classification performance.

For this purpose, we train multiple instances of a 2-block version of our energy-based model that differ in the number of iterations used when solving for the codes. For recurrent networks such as this, inference is commonly implemented by "unrolling" the network, where the parts of the network structure are repeated with parameters shared across these repeated parts to mimic an iterative algorithm that stops at a fixed number of iterations rather than at some convergence criteria.

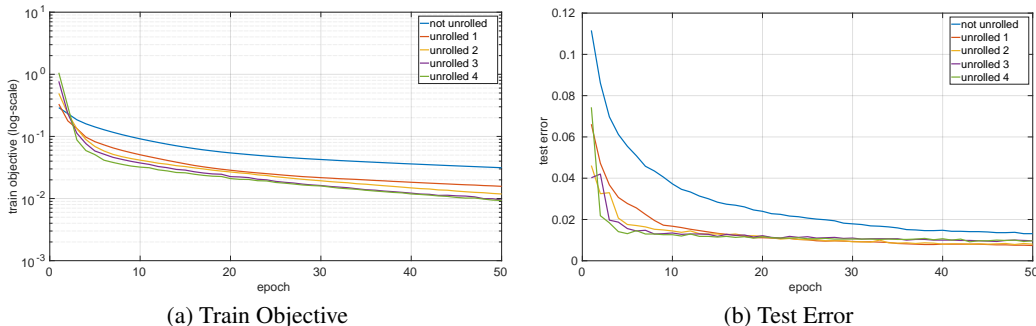

| (a) Train Objective | (b) Test Error |

Figure 3: Comparing the effects of unrolling a 2-block version of our energy-based model. (Best viewed in color.)

In Fig. 3, we compare the performance between models that were unrolled zero to four times. We see that there is a difference in performance based on how many sweeps of the variables are made. In terms of the training objective, more unrolling produces models that have lower objective values with convergence after only a few passes. In terms of testing error, however, we see that full code inference is not necessarily better, as unrolling once or twice has lower errors than unrolling three or four times. The biggest difference was between not unrolling and unrolling once, where both the training objective and testing error goes down. The testing error decreases from 0.0131 to 0.0074. While there is a clear benefit in terms of performance for unrolling at least once, there is also a trade-off between performance and computational resource, especially for deeper models.

## 4 EXPERIMENTS

We evaluate the benefits of combining top-down and bottom-up information to produce class-specific features on the CIFAR-10 (Krizhevsky & Hinton, 2009) dataset using a deep version of our EB-SSC. All experiments were performed using MatConvNet (Vedaldi & Lenc, 2015) framework with the ADAM optimizer (Kingma & Ba, 2014). The data was preprocessed and augmented following the procedure in Goodfellow et al. (2013). Specifically, the data was made zero mean and whitened, augmented with horizontal flips (with a 0.5 probability) and random cropping. No weight decay was used, but we used a dropout rate of $0.3$ before every convolution layer except for the first. For these experiments we consider a single forward pass (no unrolling).

| | Base Network | |
|---|---|---|
| block | kernel, stride, padding | activation |
| conv1 | $3 \times 3 \times 3 \times 96, 1, 1$ | ReLU/CReLU |
| conv2 | $3 \times 3 \times 96/192 \times 96, 1, 1$ | ReLU/CReLU |
| pool1 | $3 \times 3, 2, 1$ | max |
| conv3 | $3 \times 3 \times 96/192 \times 192, 1, 1$ | ReLU/CReLU |
| conv4 | $3 \times 3 \times 192/384 \times 192, 1, 1$ | ReLU/CReLU |
| conv5 | $3 \times 3 \times 192/384 \times 192, 1, 1$ | ReLU/CReLU |
| pool2 | $3 \times 3, 2, 1$ | max |
| conv6 | $3 \times 3 \times 192/384 \times 192, 1, 1$ | ReLU/CReLU |
| conv7 | $1 \times 1 \times 192/384 \times 192, 1, 1$ | ReLU/CReLU |

Table 1: Underlying block architecture common across all models we evaluated. SSC networks add an extra normalization layer after the non-linearity. And EB-SSC networks insert class-specific bias layers between the convolution layer and the non-linearity. Concatenated ReLU (CReLU) splits positive and negative activations into two separate channels rather than discarding the negative component as in the standard ReLU.

## 4.1 CLASSIFICATION

We compare our proposed EB-SSC model to that of Springenberg et al. (2015), which uses rectified linear units (ReLU) as its non-linearity. This model can be viewed as a basic feed-forward version of our proposed model which we take as a baseline. We also consider variants of the baseline model that utilize a subset of architectural features of our proposed model (e.g., concatenated rectified linear units (CReLU) and spherical normalization (SN)) to understand how subtle design changes of the network architecture affects performance.

We describe the model architecture in terms of the feature extractor and classifier. Table 1 shows the overall network architecture of feature extractors, which consist of seven convolution blocks and two pooling layers. We test two possible classifiers: a simple linear classifier (LC) and our energy-based classifier (EBC), and use softmax-loss for all models. For linear classifiers, a numerical subscript indicates which of the seven conv blocks of the feature extractor is used for classification (e.g., $LC_7$ indicates the activations out of the last conv block is fed into the linear classifier). For energy-based classifiers, a numerical subscript indicates which conv blocks of the feature extractor are replace with a energy-based classifier (e.g., $EBC_{6-7}$ indicates the activations out of conv5 is fed into the energy-based classifier and the energy-based classifier has a similar architecture to the conv blocks it replaces). The notation differ because for energy-based classifiers, the optimal activations are a function of the hypothesized class label, whereas for linear classifiers, they are not.

| Model | Train Err. (%) | Test Err. (%) | # params |
|---|---|---|---|
| ReLU+$LC_7$ | 1.20 | 11.40 | 1.3M |
| CReLU+$LC_7$ | 2.09 | 10.17 | 2.6M |
| CReLU(SN)+$LC_7$ | 0.99 | 9.74 | 2.6M |
| SSC+$LC_7$ | 0.99 | 9.77 | 2.6M |
| SSC+$EBC_{6-7}$ | 0.21 | 9.23 | 3.2M |

Table 2: Comparison of the baseline ReLU+$LC_7$ model, its derivative models, and our proposed model on CIFAR-10.

The results shown in Table 2 compare our proposed model to the baselines ReLU+$LC_7$ (Springenberg et al., 2015) and CReLU+$LC_7$ (Shang et al., 2016), and to intermediate variants. The baseline models all perform very similarly with some small reductions in error rates over the baseline CReLU+$LC_7$. However, CReLU+$LC_7$ reduces the error rate over ReLU+$LC_7$ by more than one percent (from 11.40% to 10.17%), which confirms the claims by Shang et al. (2016) and demonstrates the benefits of splitting positive and negative activations. Likewise, we see further decrease in the error rate (to 9.74%) from using spherical normalization. Though normalizing the activations doesn't add any capacity to the model, this improved performance is likely because scale-invariant activations makes training easier. On the other hand, further sparsifying the activations yielded no

benefit. We tested values $\beta = \{0.001, 0.01\}$ and found $0.001$ to perform better. Replacing the linear classifier with our energy-based classifier further decreases the error rate by another half percent (to 9.23%).

## 4.2 DECODING CLASS-SPECIFIC CODES

A unique aspect of our model is that it is generative in the sense that each layer is explicitly trying to encode the activation pattern in the prior layer. Similar to the work on deconvolutional networks built on least-squares sparse coding (Zeiler et al., 2010), we can synthesize input images from activations in our spherical coding network by performing repeated deconvolutions (transposed convolutions) back through the network. Since our model is energy based, we can further examine how the top-down information of a hypothesized class effects the intermediate activations.

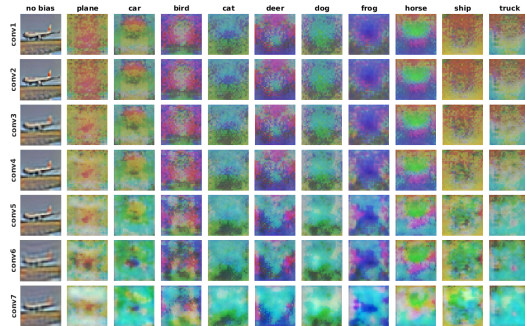

Figure 4: The reconstruction of an airplane image from different levels of the network (rows) across different hypothesized class labels (columns). The first column is pure reconstruction, i.e., unbiased by a hypothesized class label, the remaining columns show reconstructions of the learned class bias at each layer for one of ten possible CIFAR-10 class labels. (Best viewed in color.)

The first column in Fig. 4 visualizes reconstructions of a given input image based on activations from different layers of the model by convolution transpose. In this case we put in zeros for class biases (i.e., no top-down) and are able to recover high fidelity reconstructions of the input. In the remaining columns, we use the same deconvolution pass to construct input space representations of the learned classifier biases. At low levels of the feature hierarchy, these biases are spatially smooth since the receptive fields are small and there is little spatial invariance capture in the activations. At higher levels these class-conditional bias fields become more tightly localized.

Finally, in Fig. 5 we shows decodings from the conv2 and conv5 layer of the EB-SSC model for a given input under different class hypotheses. Here we subtract out the contribution of the top-down bias term in order to isolate the effect of the class conditioning on the encoding of input features. As visible in the figure, the modulation of the activations focused around particular regions of the image and the differences across class hypotheses becomes more pronounced at higher layers of the network.

## 5 CONCLUSION

We presented an energy-based sparse coding method that efficiently combines cosine similarity, convolutional sparse coding, and linear classification. Our model shows a clear mathematical connection between the activation functions used in CNNs to introduce sparsity and our cosine similarity convolutional sparse coding formulation. Our proposed model outperforms the baseline model and we show which attributes of our model contributes most to the increase in performance. We also demonstrate that our proposed model provides an interesting framework to probe the effects of class-specific coding.

## REFERENCES

Hilton Bristow, Anders Eriksson, and Simon Lucey. Fast convolutional sparse coding. In *Computer Vision and Pattern Recognition (CVPR)*, 2013.

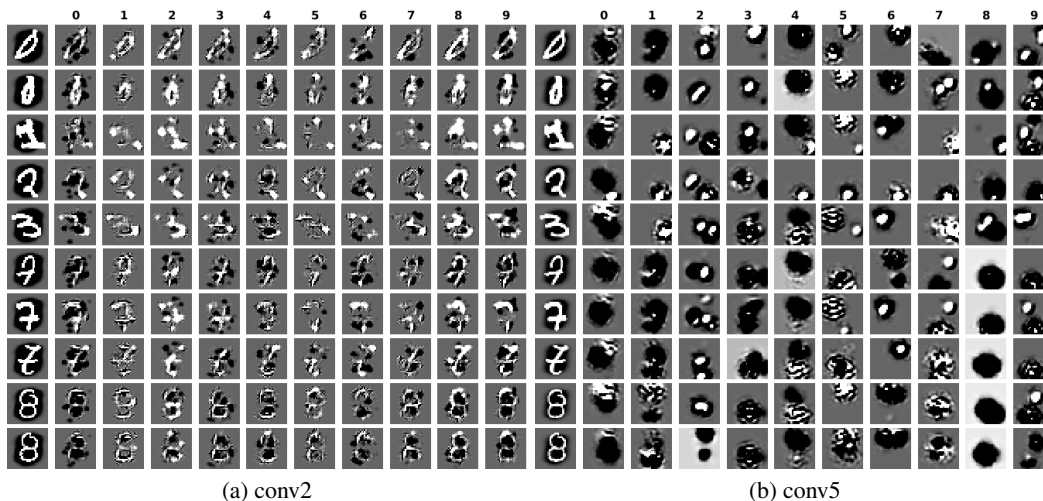

(a) conv2 (b) conv5

Figure 5: Visualizing the reconstruction of different input images (rows) for each of 10 different class hypotheses (cols) from the 2nd and 5th block activations for a model trained on MNIST digit classification.

Chunshui Cao, Xianming Liu, Yi Yang, Yinan Yu, Jiang Wang, Zilei Wang, Yongzhen Huang, Liang Wang, Chang Huang, Wei Xu, et al. Look and think twice: Capturing top-down visual attention with feedback convolutional neural networks. In *International Conference on Computer Vision (ICCV)*, 2015.

David L Donoho. Compressed sensing. *IEEE Transactions on information theory*, 2006.

Michael Elad and Michal Aharon. Image denoising via sparse and redundant representations over learned dictionaries. *IEEE Transactions on Image processing*, 2006.

Ian J Goodfellow, David Warde-Farley, Mehdi Mirza, Aaron C Courville, and Yoshua Bengio. Maxout networks. In *International conference on Machine learning (ICML)*, 2013.

Karol Gregor and Yann LeCun. Learning fast approximations of sparse coding. In *International Conference on Machine Learning (ICML)*, 2010.

Felix Heide, Wolfgang Heidrich, and Gordon Wetzstein. Fast and flexible convolutional sparse coding. In *Computer Vision and Pattern Recognition (CVPR)*, 2015.

Zhengping Ji, Wentao Huang, G. Kenyon, and L.M.A. Bettencourt. Hierarchical discriminative sparse coding via bidirectional connections. In *International Joint Converence on Neural Networks (IJCNN)*, 2011.

Zhuolin Jiang, Zhe Lin, and Larry S Davis. Learning a discriminative dictionary for sparse coding via label consistent K-SVD. In *Computer Vision and Pattern Recognition (CVPR)*, 2011.

Koray Kavukcuoglu, Pierre Sermanet, Y-Lan Boureau, Karol Gregor, Michaël Mathieu, and Yann L Cun. Learning convolutional feature hierarchies for visual recognition. In *Advances in neural information processing systems (NIPS)*, 2010.

Diederik Kingma and Jimmy Ba. Adam: A method for stochastic optimization. *arXiv preprint arXiv:1412.6980*, 2014.

Alex Krizhevsky and Geoffrey Hinton. Learning multiple layers of features from tiny images. 2009.

Hugo Larochelle and Yoshua Bengio. Classification using discriminative restricted boltzmann machines. In *International conference on Machine learning (ICML)*, 2008.

Yann LeCun, Sumit Chopra, Raia Hadsell, M Ranzato, and F Huang. A tutorial on energy-based learning. *Predicting structured data*, 2006.

Xin Li and Yuhong Guo. Bi-directional representation learning for multi-label classification. In *Joint European Conference on Machine Learning and Knowledge Discovery in Databases (ECML KDD)*. 2014.

Bruno A Olshausen and David J Field. Sparse coding with an overcomplete basis set: A strategy employed by v1? *Vision research*, 1997.

Xiaofeng Ren and Deva Ramanan. Histograms of sparse codes for object detection. In *Computer Vision and Pattern Recognition (CVPR)*, 2013.

Christopher J Rozell, Don H Johnson, Richard G Baraniuk, and Bruno A Olshausen. Sparse coding via thresholding and local competition in neural circuits. *Neural computation*, 2008.

Wenling Shang, Kihyuk Sohn, Diogo Almeida, and Honglak Lee. Understanding and improving convolutional neural networks via concatenated rectified linear units. In *International conference on Machine learning (ICML)*, 2016.

J Springenberg, Alexey Dosovitskiy, Thomas Brox, and M Riedmiller. Striving for simplicity: The all convolutional net. In *International conference on Learning Representations (ICLR) (workshop track)*, 2015.

Mark Tygert, Arthur Szlam, Soumith Chintala, Marc'Aurelio Ranzato, Yuandong Tian, and Wojciech Zaremba. Convolutional networks and learning invariant to homogeneous multiplicative scalings. *arXiv preprint arXiv:1506.08230*, 2015.

A. Vedaldi and K. Lenc. Matconvnet – convolutional neural networks for matlab. In *ACM International Conference on Multimedia*, 2015.

John Wright, Allen Y Yang, Arvind Ganesh, S Shankar Sastry, and Yi Ma. Robust face recognition via sparse representation. *IEEE transactions on pattern analysis and machine intelligence (TPAMI)*, 2009.

Allen Y Yang, Zihan Zhou, Arvind Ganesh Balasubramanian, S Shankar Sastry, and Yi Ma. Fast-minimization algorithms for robust face recognition. *IEEE Transactions on Image Processing*, 2013.

Jianchao Yang, Kai Yu, and Thomas Huang. Supervised translation-invariant sparse coding. In *Computer Vision and Pattern Recognition (CVPR)*, 2010.

Matthew D. Zeiler, Dilip Krishnan, Graham W. Taylor, and Robert Fergus. Deconvolutional networks. In *Computer Vision and Pattern Recognition (CVPR)*, 2010.

Yangmuzi Zhang, Zhuolin Jiang, and Larry S Davis. Discriminative tensor sparse coding for image classification. In *British Machine Vision Conference (BMVC)*, 2013.

Ning Zhou, Yi Shen, Jinye Peng, and Jianping Fan. Learning inter-related visual dictionary for object recognition. In *Computer Vision and Pattern Recognition (CVPR)*, 2012.

## APPENDIX A

Here we show that spherical sparse coding (SSC) with a norm constraint on the reconstruction is equivalent to standard convolutional sparse coding (CSC). Expanding the least squares reconstruction error and dropping the constant term $\|x\|^2$ gives the CSC problem:

$$\max_{\mathbf{z}} \ 2\mathbf{x}^{\mathsf{T}}\big(\sum_{k=1}^{K}\mathbf{d}_k * \mathbf{z}_k\big) - \|\sum_{k=1}^{K}\mathbf{d}_k * \mathbf{z}_k\|_2^2 - \beta\sum_{k=1}^{K}\|\mathbf{z}_k\|_1.$$

Let $\epsilon = \|\sum_{k=1}^{K}\mathbf{d}_k * \mathbf{z}_k\|_2$ be the norm of the reconstruction for some code $\mathbf{z}$ and let $\mathbf{u}$ be the reconstruction scaled $\epsilon$ to have unit norm so that:

$$\mathbf{u} = \frac{\sum_{k=1}^{K}\mathbf{d}_k * \mathbf{z}_k}{\|\sum_{k=1}^{K}\mathbf{d}_k * \mathbf{z}_k\|_2} = \sum_{k=1}^{K}\mathbf{d}_k * \bar{\mathbf{z}}_k \quad \text{with} \quad \bar{\mathbf{z}} = \frac{1}{\epsilon}\mathbf{z}$$

We rewrite the least-squares objective in terms of these new variables:

$$\max_{\bar{\mathbf{z}},\epsilon>0} g(\bar{\mathbf{z}},\epsilon) = \max_{\bar{\mathbf{z}},\epsilon>0} 2\mathbf{x}^{\mathsf{T}}\big(\epsilon\mathbf{u}\big) - \|\epsilon\mathbf{u}\|_2^2 - \beta\|\epsilon\bar{\mathbf{z}}\|_1$$

$$= \max_{\bar{\mathbf{z}},\epsilon>0} 2\epsilon\big(\mathbf{x}^{\mathsf{T}}\mathbf{u} - \frac{\beta}{2}\|\bar{\mathbf{z}}\|_1\big) - \epsilon^2$$

Taking the derivative of $g$ w.r.t. $\epsilon$ yields the optimal scaling $\epsilon^*$ as a function of $\bar{\mathbf{z}}$:

$$\epsilon(\bar{\mathbf{z}})^* = \mathbf{x}^{\mathsf{T}}\mathbf{u} - \frac{\beta}{2}\|\bar{\mathbf{z}}\|_1.$$

Plugging $\epsilon(\bar{\mathbf{z}})^*$ back into $g$ yields:

$$\max_{\bar{\mathbf{z}},\epsilon>0} g(\bar{\mathbf{z}},\epsilon) = \max_{\bar{\mathbf{z}},\|u\|_2=1} \big(\mathbf{x}^{\mathsf{T}}\mathbf{u} - \frac{\beta}{2}\|\bar{\mathbf{z}}\|_1\big)^2.$$

Discarding solutions with $\epsilon < 0$ can be achieved by simply dropping the square which results in the final constrained problem:

$$\arg\max_{\bar{\mathbf{z}}} \mathbf{x}^{\mathsf{T}}\big(\sum_{k=1}^{K}\mathbf{d}_k * \bar{\mathbf{z}}_k\big) - \frac{\beta}{2}\sum_{k=1}^{K}\|\bar{\mathbf{z}}_k\|_1$$

$$\text{s.t.} \quad \|\sum_{k=1}^{K}\mathbf{d}_k * \bar{\mathbf{z}}_k\|_2 \le 1.$$

## APPENDIX B

We show in this section that coding in the EB-SSC model can be solved efficiently by a combination of convolution, shrinkage and projection, steps which can be implemented with standard libraries on a GPU. For convenience, we first rewrite the objective in terms of cross-correlation rather than convolution (i.e., , $\mathbf{x}^{\mathsf{T}}(\mathbf{d}_k * \mathbf{z}_k) = (\mathbf{d}_k \star \mathbf{x})^{\mathsf{T}}\mathbf{z}_k$). For ease of understanding, we first consider the coding problem when there is no classification term.

$$\mathbf{z}^* = \arg\max_{\|\mathbf{z}\|_2^2 \le 1} \mathbf{v}^{\mathsf{T}}\mathbf{z} - \beta\|\mathbf{z}\|_1,$$

where $\mathbf{v} = [(\mathbf{d}_1 \star \mathbf{x})^{\mathsf{T}},\ldots,(\mathbf{d}_K \star \mathbf{x})^{\mathsf{T}}]^{\mathsf{T}}$. Pulling the constraint into the objective, we get its Lagrangian function:

$$\mathcal{L}(\mathbf{z},\lambda) = \mathbf{v}^{\mathsf{T}}\mathbf{z} - \beta\|\mathbf{z}\|_1 + \lambda\big(1 - \|\mathbf{z}\|_2^2\big).$$

From the partial subderivative of the Lagrangian w.r.t. $z_i$ we derive the optimal solution as a function of $\lambda$; and from that find the conditions in which the solutions hold, giving us:

$$z_i(\lambda)^* = \frac{1}{2\lambda} \cdot \begin{cases} v_i - \beta & v_i > \beta \\ 0 & \text{otherwise} \\ v_i + \beta & v_i < \beta \end{cases}. \tag{17}$$

This can also be compactly written as:

$$\mathbf{z}(\lambda)^* = \frac{1}{2\lambda}\tilde{\mathbf{z}}, \tag{18}$$

$$\tilde{\mathbf{z}} = \mathbf{s}^2 \odot \mathbf{v} - \beta\mathbf{s}$$

where $\mathbf{s} = \text{sign}(\mathbf{z}^*) \in \{-1, 0, 1\}^{|\mathbf{z}|}$ and $\mathbf{s}^2 = \mathbf{s} \odot \mathbf{s} \in \{0, 1\}^{|\mathbf{z}|}$. The sign vector of $\mathbf{z}^*$ can be determined without knowing $\lambda$, as $\lambda$ is a Lagrangian multiplier for an inequality it must be non-negative and therefore does not change the sign of the optimal solution. Lastly, we define the squared $\ell_2$-norm of $\tilde{\mathbf{z}}$, a result that will be used later:

$$\begin{aligned}\|\tilde{\mathbf{z}}\|_2^2 &= \tilde{\mathbf{z}}^{\mathsf{T}}(\mathbf{s}^2 \odot \mathbf{v}) - \beta\tilde{\mathbf{z}}^{\mathsf{T}}\mathbf{s} \\ &= \tilde{\mathbf{z}}^{\mathsf{T}}\mathbf{v} - \beta\|\tilde{\mathbf{z}}\|_1.\end{aligned} \tag{19}$$

Substituting $\mathbf{z}(\lambda)^*$ back into the Lagrangian we get:

$$\mathcal{L}(\mathbf{z}(\lambda)^*, \lambda) = \frac{1}{2\lambda}\mathbf{v}^{\mathsf{T}}\tilde{\mathbf{z}} - \frac{\beta}{2\lambda}\|\tilde{\mathbf{z}}\|_1 + \lambda\Big(1 - \frac{1}{4\lambda^2}\|\tilde{\mathbf{z}}\|_2^2\Big),$$

and the derivative w.r.t. $\lambda$ is:

$$\frac{\partial\mathcal{L}(\mathbf{z}(\lambda)^*}{\partial\lambda} = -\frac{1}{2\lambda^2}\mathbf{v}^{\mathsf{T}}\tilde{\mathbf{z}} + \frac{\beta}{2\lambda^2}\|\tilde{\mathbf{z}}\|_1 + 1 + \frac{1}{4\lambda^2}\|\tilde{\mathbf{z}}\|_2^2.$$

Setting the derivative equal to zero and using the result from Eq. 19, we can find the optimal solution to $\lambda$:

$$\lambda^2 = \frac{1}{2}\tilde{\mathbf{z}}^{\mathsf{T}}\mathbf{v} - \frac{\beta}{2}\|\tilde{\mathbf{z}}\|_1 - \frac{1}{4}\|\tilde{\mathbf{z}}\|_2^2 = \frac{1}{2}\|\tilde{\mathbf{z}}\|_2^2 - \frac{1}{4}\|\tilde{\mathbf{z}}\|_2^2$$

$$\implies \lambda^* = \frac{1}{2}\|\tilde{\mathbf{z}}\|_2.$$

Finally, plugging $\lambda^*$ into Eq. 18 we find the optimal solution

$$\mathbf{z}^* = \frac{\tilde{\mathbf{z}}}{\|\tilde{\mathbf{z}}\|_2}. \tag{20}$$

