# Peer review of "Energy-Based Spherical Sparse Coding"

_ICLR 2017 — rejected_

[Official Review · AnonReviewer2 · rating 6 · confidence 4 · 16 Dec 2016]

First, I'd like to thank the authors for their answers and clarifications.
I find, the presentation of the multi-stage version of the model much clearer now.

Pros:

+ The paper states a sparse coding problem using cosine loss, which allows to solve the problem in a single pass.

+ The energy-based formulation allows bi-directional coding that incorporates top-down and bottom-up information in the feature extraction process. 

Cons:

+ The cost of running the evaluation could be large in the  multi-class setting, rendering the approach less attractive and the computational cost comparable to recurrent architectures.

+ While the model is competitive and improves over the baseline, the paper would be more convincing with other comparisons (see text). The experimental evaluation is limited (a single database and a single baseline)

------

The motivation of the sparse coding scheme is to perform inference in a feed forward manner. This property does not hold in the multi stage setting, thus optimization would be required (as clarified by the authors).

Having an efficient way of performing a bi-directional coding scheme is very interesting. As the authors clarified, this could not necessarily be the case, as the model needs to be evaluated many times for performing a classification.

Maybe an interesting combination would be to run the model without any class-specific bias, and evaluation only the top K predictions with the energy-based setting.

Having said this, it would be good to include a discussion (if not direct comparisons) of the trade-offs of using a model as the one proposed by Cao et al. Eg. computational costs, performance.

Using the bidirectional coding only on the top layers seems reasonable: one can get a good low level representation in a class agnostic way. This, however could be studied in more detail, for instance showing empirically the trade offs. If I understand correctly, now only one setting is being reported.

Finally, the authors mention that one benefit of using the architecture derived from the proposed coding method is the spherical normalization scheme, which can lead to smoother optimization dynamics. Does the baseline (or model) use batch-normalization? If not, seems relevant to test.


Minor comments:

I find figure 2 (d) confusing. I would not plot this setting as it does not lead to a function (as the authors state in the text).

[Official Review · AnonReviewer3 · rating 5 · confidence 4 · 18 Dec 2016]
**No Title**

This paper proposes sparse coding problem with cosine-loss and integrated it as a feed-forward layer in a neural network as an energy based learning approach. The bi-directional extension makes the proximal operator equivalent to a certain non-linearity (CReLu, although unnecessary). The experiments do not show significant improvement against baselines. 

Pros: 
- Minimizing the cosine-distance seems useful in many settings where compute inner-product between features are required. 
- The findings that the bidirectional sparse coding is corresponding to a feed-forward net with CReLu non-linearity. 

Cons:
- Unrolling sparse coding inference as a feed-foward network is not new. 
- The class-wise encoding makes the algorithm unpractical in multi-class cases, due to the requirement of sparse coding net for each class. 
- It does not show the proposed method could outperform baseslines in real-world tasks.

[Official Review · AnonReviewer1 · rating 5 · confidence 4 · 21 Dec 2016]

The paper introduces an efficient variant of sparse coding and uses it as a building block in CNNs for image classification. The coding method incorporates both the input signal reconstruction objective as well as top down information from a class label. The proposed block is evaluated against the recently proposed CReLU activation block.

Positives:
The proposed method seems technically sound, and it introduces a new way to efficiently train a CNN layer-wise by combining reconstruction and discriminative objectives.

Negatives:
The performance gain (in terms of classification accuracy) over the previous state-of-the-art is not clear. Using only one dataset (CIFAR-10), the proposed method performs slightly better than the CRelu baseline, but the improvement is quite small (0.5% in the test set). 

The paper can be strengthened if the authors can demonstrate that the proposed method can be generally applicable to various CNN architectures and datasets with clear and consistent performance gains over strong CNN baselines. Without such results, the practical significance of this work seems unclear.

[Final Decision · Program Chairs · 06 Feb 2017]
**ICLR committee final decision**

This paper proposes a variant of convolutional sparse coding with unit norm code vectors using cosine distance to evaluate reconstructions. The performance gains over baseline networks are quite minimal and demonstrated on limited datasets, therefore this work fails to demonstrate practical usefulness, while the novelty of the contribution is too slight to stand on its own merit.